# The Relationship between Dividend Policy and Earnings Quality: The Role of Accounting Information in Indonesia's Capital Market

Muljanto Siladjaja [1], Yuli Anwar [2,*] and Ismulyana Djan [2]

[1]  Faculty of Economics and Business, IKPIA Perbanas Institute, Jakarta 12940, Indonesia; muljanto.siladjaja@perbanas.id
[2]  Faculty of Economics and Business, Universitas Binaniaga Indonesia, Bogor 16128, Indonesia; ismulyanadjan@unbin.ac.id
[*]  Correspondence: julianwar@unbin.ac.id

**Abstract:** This study provides empirical proof that the positive impact of high accrual quality is the ability to accurately predict the future return with a positive sign. In the capital market, better prospects are commonly indicated by regularly and routinely implementing a high-yield dividend policy. This study uses dividend policy as a moderated multiple regression, which plays a critical role in achieving a high obedience to accounting standards. The causal research involved 154 of the companies listed on the Indonesia Capital Market and 384 observations in the industrial manufacturing sector for 2015–2020. By mulling over the effect of the COVID-19 pandemic in 2020, and predicting the future market using zero growth with no assumed growth in the future, this empirical study shows that dividend policy is critical when minimizing opportunity behavior. This research provides a mapping of the decision tree model as an implication of game theory because of the interactive feedback between the earning quality and future market value. A sign such as "good" news significantly stimulates the perception of optimistic investors, with no negative manipulation and accruals. It paves the way for investors to strictly control and monitor strategic decisions to obtain significant improvement in prospects.

**Keywords:** discretionary accrual quality; tax management; high-yield dividend; future market

## 1. Introduction

Future return is a primary indicator of a high rate of concern, whereby an investor emphasizes a high earning quality, which reflects the volatile fluctuation of risk during the publication period (Savor and Wilson 2016). High-quality financial reporting positively impacts the perception of investors (Pompili and Tutino 2019); when it illustrates actual earnings, it is related to the existence of this investment in a secured area. High-quality accounting information paves the way to accurately predict future returns, including capital costs. High earning quality is a sign indicating whether management performance is on the right track (Lebert 2019; Pompili and Tutino 2019) linked to the phenomenon of opportunity behavior (Kothari et al. 2016; Li 2019). Furthermore, Dichev et al. (2016) pointed out that low accrual quality was misleading information when publishing a financial report; thus, improving accrual quality has a positive effort on the quality of accounting information. Therefore, this research proves that earnings quality is a high accuracy indicator of illustrating real earnings as an efficient contracting, backed by no distortion.

The current investor in this empirical study is a strategic investor who pays attention to financial reporting quality (Mehrani et al. 2017; Hoang et al. 2019). The patterns of dividend policy in Indonesia (Figure 1) reveal that the average rate of dividend payout is 31.33%, while the average growth market price is 26.75%, indicating a high and periodic implementation of the dividend payout. By learning about the high growth market price as attractive

investing in the Indonesia Capital Market, this "good news" encourages this research to focus on the capability of this market to reach a better return in the previous period. Because of the excellent performance of this capital market, this empirical literature depicts the capability of predicting the future return precisely based on accounting information. This phenomenon of dividend policy has been investigated by (Kasanen et al. 1996) when the management's effort for realizing the expected return has been the most critical factor for opportunistic behavior. It illustrated how the Finland companies had to adjust the earnings for funding this dividend policy because of disseminating the companies on the right track. Kato et al. (2002) identified the same phenomenon in Japanese banks, which was used to smooth the intensity of agency conflict because of solid management willingness as a signal of the company being on the right track, related to the trust of investors. This one has been supported by (Shah et al. 2010) and underlined that Pakistani and Chinese Companies have the same treatment model when the dividend is a sign of minimum opportunistic behavior. Concerning the comparative case for the effect of dividend policy in Indonesia, it has been proved that the dividend and accounting treatment policy have related to each other, which depends on reaching a high sustainability of business existence in the long run. Partially, the dividend policy is related to the cost of capital as an indicator of risk; it can be generalized that this policy has a positive impact on the performance in the following period. As one consideration in stimulating the positive movement of market share, there is a firm willingness to implement this policy with high growth. This high-yield dividend policy led to a reduction in internal conflict at the minimum level (Chansarn and Chansarn 2016; He et al. 2017; Deng et al. 2017). This is a debatable issue in finance; however, this policy has the advantage of resulting in a low capital cost (Wei He and Kyaw 2018). This study provides a comprehensive illustration of the high-yield dividend policy, sales, and retained earnings (Figure 1).

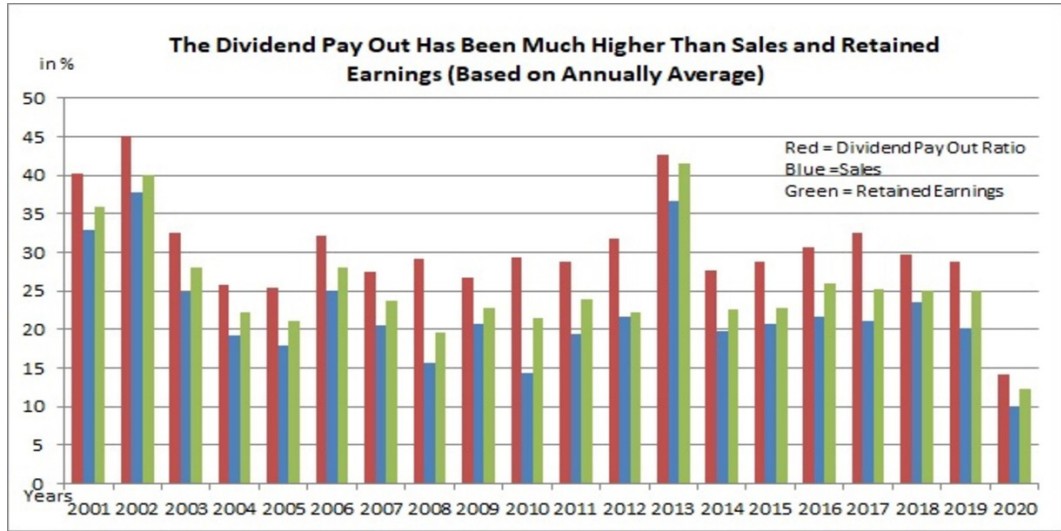

**Figure 1.** The pattern of high-yield dividends from 2001–2020; COVID-19 was declared a pandemic in February 2020. Source: This compiled data from the Annual Report of Indonesia Stock Exchange 2020 (IDX, ISE 2001–2020).

According to the literature (Wu et al. 2016), high earning quality has a positive correlation with business survival, as supported by the study of (Datta et al. 2013), which related it to the dividend policy. This study presented the impact of accrual quality on the estimated market price in the future, including a predictive model for estimating the probability of investor action using Bayes' theorem. By combining the phenomena of accrual quality and dividend policy, Pathak and Ranajee (2020) and (Siladjaja and Anwar 2020) identified accruals and dividends as indicators of the management being on the right track. Because of misleading information, high-quality accounting information is the main factor responsible

for positively forming investor perception, which indicates that a higher accrual quality leads to higher prospects.

The structural advantages of accounting information are to provide the investor with the protective right for the positive expected return. When the accurate information of current performance illustrates none of the misleading information due to agency theory, it refers to a guarantee of high preferences investment indicated by the high earnings quality. By having high investors' trust, its incentive for management to level up the efficient rate and production capacity positively impacts the growth indicator of macroeconomics; this one has linked the stimulating effect of accounting information on gaining the actual economic performance (Eldomiaty et al. 2020; Wang and Li 2020). The high earnings quality is a sustainable issue pragmatically; when there is a high-precision method to calculate the accrual quality as a proxy of obedience to accounting standards and compliance with the tax regulation. Meanwhile, it should be followed up by the regulators to fix high-quality financial reporting as the mandatory obligation for coping with the volatile movements of market price and keeping the high investor's trust„ including the dividends as an empirical fact of efficient contracting.

This study empirically determines the role of the accounting standard and tax regulations at a maximum level in lowering risk, with a positive contribution to the future market value as an indicator of future return. Practically, management is willing to make a high tax payment to maximize firm value, which includes deducting taxable income (Ifada and Wulandari 2015; Liu and Lee 2019). A violation of tax regulations is unacceptable; thus, investors have to be aware of any volatile movement of agency costs in the future (Lee 2016). Moreover, Kałdoński and Jewartowski (2020) and Jacob and Schütt (2020) declared that high obedience to tax regulations is critical for the optimistic perception of investors because it is correlated with a high probability of tax investigation in the following period (Hu et al. 2015). In the same vein, Duy and Tran (2020) identified a negative contribution of accounting accruals to tax accruals, whereby high-quality financial reporting could improve compliance with tax regulations. Empirically, this study proves how accrual quality and tax management positively contribute to a future return. The role of dividend policy in moderating the linkage of both variables to future market value is revealed through a mapping model for investor action using Bayes' theorem (Hutton and Stocken 2021). The research objective was to test the relationship between accrual quality and tax management concerning future market value using dividend policy as a moderating variable, as well as to depict the interactive feedback between investor perception and accounting information. The contribution of this research is to stress the meaning of high-quality accounting information as a piece of high-precision information for distinguishing a better prospect or not when it is related to a considerable effort to prohibit the Peach–Lemon Thesis (Akerlof 1970). Undeniably, it paves the way for investors to monitor the existing business' high sustainability; high obedience and compliance have been essential prerequisites as the items of predictability in generating a high expected return.

## 2. Theoretical Framework and Hypothesis

When dividend policy has stimulated the positive movement of market price, it is used to disseminate the available signal as a communication process to grab a high investor's trust. Because of the implication of agency theory, there is a gap in asymmetries information related to the dysfunctional behavior of management in inserting a distortion (Dichev et al. 2016). Based on (Kothari 2001), this phenomenon can be explicitly explained by the positive accounting theory (PAT). Firstly, this empirical explanation, stated by Watts and Zimmerman (2003), explains that accounting policies are a sensitive discussion between the investor and management in publishing a financial report. Management has high flexibility in determining the model of accounting policies, with the best choice giving an advantage in calculating the current period's earnings. This theory provides a solution for the classic phenomena in a capital market, such as the fluctuation of market price; it is assumed that

a company represents a nexus of contracts (Scott 2016), where the accrual quality plays a critical role in efficiently constructing an opportunistic motive according to agency theory.

The basis of this theory is the capability of the management and investors to assess information due to its asymmetry. Management has the dominant information, which represents a conflict of interest. With the responsibility of managing firm assets, management uses financial reports to share valid information, including the communication process, thereby sending a signal. An investor can judge this information as a positive or negative signal when calculating the expected future return. Volatile market price fluctuation is a response from an investor to low-quality financial reporting, where there is no investor protection; thus, this refers to a high investment risk (Jeong and Sohn 2013; Persakis and Iatridis 2017; Ezat 2019).

During the rapid growth of investment in the capital market, the regulator should formally guide management in designing the accounting policy; mainly, the regulator aims to improve the investor's trust by proposing a safe area. By implementing corporate governance as an anticipative solution against the Sarbanes Oxley Act (2003), management has an ethical liability to increase the level of transparency in decision making. This theory states that the government's involvement has practical implications in determining actual earnings, including fiscal and monetary policies. *Regulatory capture theory* (Godfrey et al. 2014) can be used to reflect how an investor motivates the high compliance of management. Because of the gap between accounting and tax regulations, the majority shareholder tolerates different taxable income calculations as a fiscal correction (Stigler 2012). A negative consequence of low compliance with tax regulations directly impacts the high agency cost in the following period (Yorke et al. 2016; Liu and Lee 2019).

Concerning the Agency Theory and Positive Accounting Theory, this pressure of high-quality financial reporting has been triggered by dividend policy as a guideline of signaling effect. Empirically, the benefit of this policy is to sign this firm with low risk and high probability to reach a better prospect, and this one means the capability of an external funding party. It refers to an external third party; the role of the tax regulator as a fiscal authority impacts the agency cost in the subsequent period, based on the regulation capture theory (Stigler 2012). This one drives the investor to be alert to the possibility of tax investigation because (Elayan et al. 2016) and (Alipour et al. 2019) stress the role of high earnings quality on the firm value.

The characteristics of the investor in this research, who is concerned with the high earnings quality, has a rational decision model because (Mehrani et al. 2017) and (Sakaki et al. 2021) proved the pattern of institutional investor as an assumption of efficient market hypothesis. By linking the earnings quality and rational decision model, this research has tested the relationship of the impact of high-quality accounting information on the investor's decision in the maximum of each's utility as proof of the implication of game theory. This causal relationship has been supported by (Chen and Wu 2021), underlining the investor's decision to take a short position when the earnings quality is low. They stressed the negative investor perception of high tax avoidance as low compliance and volatile tax agency in the future. The mutual relationship encourages the research to modify the decision model based on (Kaplan 1996) and (Askari et al. 2019) by combining the decision tree model and Bayes' theorem to predict the probability of investor action.

## 2.1. Hypothesis Development

When the firm has a high earning quality, another party cannot take advantage of the abnormal return (Ping 2016; El Diri et al. 2020) because it illustrates the actual earnings in the current period. Thus, there is a direct positive contribution of earning quality to the firm's value. Jeong and Choi (2019) found that persistent earnings have a positive impact on future earnings; this was proven by Ozili (2016) and Takacs et al. (2020) as a function of the positive movement of market price during the publication period. As a novelty in estimating the fluctuation of market price in the future period, the central hypothesis can be arranged systematically as follows:

**Hypothesis of Testing Accrual Quality.** *The accrual quality has a positive influence on future performance.*

According to Dichev et al. (2016) and Beyer et al. (2019), high earning quality is a good signal because it indicates the ability of the firm to attain better performance in the future. High accrual quality has some advantages when calculating future returns with high accuracy because opportunistic behavior is at a minimum (Al-Rassas and Kamardin 2016; Park et al. 2021). For the implication of accruals in determining total netbook assets and earning value, this study distinguishes future performance as a function of equity and earnings. Firstly, a comparison between future price and current book value is proposed, indicating the effect of future market value on equity. Secondly, the future price and current earnings are compared, indicating the effect of future market value on earnings. The following hypotheses can be extended from the first hypothesis:

**Hypothesis H1.** *Discretionary accrual quality positively influences future market value as a function of equity.*

**Hypothesis H2.** *Discretionary accrual quality positively influences future market value as a function of earnings.*

When considering tax avoidance (Desai and Dharmapala 2005; Salihu et al. 2013), management has the proclivity to deduct taxable income as tax savings. This is supported by Lennox et al. (2013), Lee (2016), and Liu and Lee (2019), who measured high tax conformity using discretionary tax accrual quality as an indicator of high compliance. Choudhary et al. (2016) pointed out that a positive relationship between tax management and earning quality represents opportunistic behavior when calculating actual earnings. This is supported by Jacob and Schütt (2020), who introduced the term "corporate tax avoidance". Báez-Díaz and Alam (2012) and Lee (2016) underlined that all efforts to infringe on tax regulations have negative consequences on future earnings due to the high probability of tax investigation. The final impact is high agency cost, as supported by Liu and Lee (2019). As a new model for estimating the impact of tax accrual quality on the fluctuation of future market price, the second hypothesis can be systematically proposed as follows:

**Hypothesis of Testing Tax Management.** *Tax management positively influences future performance.*

A higher level of tax management indicates high compliance with tax regulations at the maximum level. It describes the actual performance in reporting during the current period. By separating the implications of tax management on equity and earnings, the second hypothesis can be extended as follows:

**Hypothesis H3.** *Discretionary tax accrual quality positively influences future market value as a function of equity.*

**Hypothesis H4.** *Discretionary tax accrual quality positively influences future market value as a function of earnings.*

By examining the benefits of dividend policy as a signaling effect, Chaudhary et al. (2016), Chansarn and Chansarn (2016), and Nekhili et al. (2016) revealed that management exhibits a solid willingness to implement a dividend policy, reflecting a sacrifice to achieve better prospects. Implementing a high-dividend policy indicates real and actual current earnings, in addition to stressing the positive correlation between earning quality and prospects (Mongrut and Winkelried 2019; Orazalin and Akhmetzhanov 2019). In addition, Pathak and Ranajee (2020) explained that the dividend policy is a critical indicator of better

performance in the future, related to expected return. Therefore, the third hypothesis can be proposed as follows:

**Hypothesis of Testing on Dividend and Accrual Quality.** *Dividend policy strengthens accrual quality on future performance.*

Accruals have a certain impact on equity and earnings; thus, the third hypothesis can be extended as follows:

**Hypothesis H5.** *Dividend policy strengthens the positive impact of discretionary accrual quality on future market value as a function of equity.*

**Hypothesis H6.** *Dividend policy strengthens the positive impact of discretionary accrual quality on future market value as a function of earnings.*

Lennox et al. (2013) and Yorke et al. (2016) found a negative relationship between book and tax accruals; therefore, this study investigates how tax management significantly contributes to the perception of optimistic investors using dividend policy as a moderating variable. This is a proxy for estimating the impact of dividend policy on tax management concerning future performance. Thus, the fourth hypothesis can be proposed as follows:

**Hypothesis of Testing on Dividend and Tax Management.** *Dividend policy strengthens the impact of tax management on future performance.*

Miiller and Martinez (2016) pointed out that investors appreciate compliance with the tax regulations, and they revealed the gap between accounting standards and tax regulations in a limited tolerance zone, thereby correctly anticipating future fiscal correction by determining taxable income. By considering an opportunistic motive in tax management, Tang and Firth (2011), Báez-Díaz and Alam (2012), and Choudhary et al. (2016) tested how dividend policy contributes to tax management in achieving better prospects, categorizing the statistical testing of future market value into equity and earnings. Thus, the fourth hypothesis can be systematically proposed as follows:

**Hypothesis H7.** *Dividend policy strengthens the positive impact of discretionary tax accruals on future market value as a function of equity.*

**Hypothesis H8.** *Dividend policy strengthens the positive impact of discretionary tax accruals on future market value as a function of earnings.*

*2.2. The Empirical Conceptual Research Framework*

The research framework is organized in Figure 2.

The research framework in Figure 2 indicates that causal research was used to test the impact of earning quality and tax management on future market value as a proxy for prospect measurement. This was moderated by dividend policy, representing the phenomenon of volatile movement in market price. To increase the validity of this paper, some control variables were introduced to indicate the significant contribution of other variables to future market value. These control variables played a crucial role in measuring accounting quality, referring to opportunistic behavior when publishing annual reports, which can be signposted by a low level of bias with minimum error. Therefore, the contribution of control variables could be determined using the regression model to measure the impact of external factors on prospects. Dividend policy was statistically tested as a moderating variable considering the effect of a high-yield dividend policy. This allows for detecting management's willingness to smooth the intensity of internal conflict, thus obtaining a low capital cost, indicating a high probability of fulfilling the expected return in the following period.

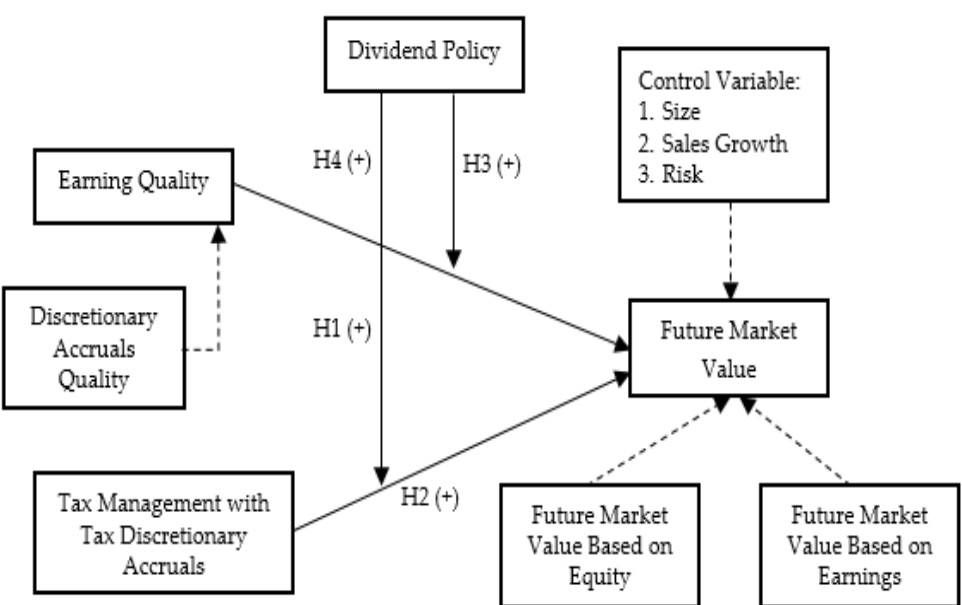

**Figure 2.** The empirical conceptual research framework.

## 3. Method

### 3.1. Sample and Data Collection

This qualitative study implemented a causal method using purposive sampling; all observation data were collected for the period 2007–2020 from manufacturing industries listed on the Indonesia Capital Market. This model aims to collect the high-quality data as a representative sampling, where each sample has the same probability of being a chosen object for research. Based on the population data, it is expected to take a typical generalized result concerning the companies that have periodically implemented dividend policy. It is related to designing some criteria in obtaining a high validity. The inclusion criteria were as follows (Sekaran and Bougie 2016): (1) the company made dividend payments within the observation period; (2) the company had a positive average annual growth rate. This empirical study gathered 384 observations from 154 companies, using secondary data obtained from the ICMD (Indonesia Market Capital Directory), the Indonesia Stock Exchange (www.idx.co.id), and Yahoo Finance.

By measuring beta as a proxy of measurement risk based on CAPM, this calculation of market price fluctuations uses the geometry approach, when (Damodaran 2012) uses this model to obtain the average number. This average treatment has been implemented to calculate the growth rate during the observation period, where it assumes the number in the beginning and end period based on the annually periodic have the same probability. All measurements of all variables have been measured by ratio where it depicts the measurement are real, relative size, or measurable indicators. By testing the impact of accounting information on the fluctuation of market price in the future, this predictive model uses multiple regression with the moderated variable, which was inspired by (Perotti and Wagenhofer 2014) by using the residual error value as an indicator of earnings quality and (Nekhili et al. 2016) with using the dividend policy as a moderated variable. The uniqueness of measurement earnings quality is to multiply the discretionary accrual with −1 (minus one); it can be interpreted that the substantial effort is to have high obedience and compliance with the accessible regulation legally to reach the lowest misleading information so that a small error coefficient can indicate it in a regression formula. The same treatment is performed to measure tax accrual quality. The use of regression in testing the measurement of earnings quality has been supported by (Pompili and Tutino 2019) and (Lebert 2019).

*3.2. Measurement*

3.2.1. Future Market Value

Future Market Value is an estimated return in the future period, which is the period for t + 1. Briefly, it can be interpreted that the market value has been predicted. As a unique indicator for the future return, this measurement depicted the characteristics of this future market value, which can be broken down as below:

1.  Future Market Value is a fact-based and objective indicator of "good news", an accurately comparative analysis of the current and predicted market prices in the following period. It is related to calculating the risk and cost of capital due to the high probability of reaching a better prospect.
2.  The Future Market Value has been a trustworthy sign of both over- and undervaluation when the high complexity of accounting information has been an obstacle in analysing the capability of management in keeping a high sustainability in the long run. As a need of a simple indicator of management viewpoint, the dividend policy can be used to estimate the firm value, including a proxy for high obedience and compliance.
3.  The Future Market Value indicates the real risk and reasonable cost of capital when it refers to the present value of the predicted market price in the following period. By adjusting the present value of the dividend and market price with the expected return, this indicator reflects an accurate illustration in estimating the future return, particularly in overcoming the market risk, which it adopted from the CAPM model.

To new literature, the Future Market Value is an innovative approach to illustrating the investor's perception based on the capability of management performance in fulfilling the targeted return. Anticipating the pandemic's effect in 2020–2022 (Damodaran 2012), the estimated price with the H model was calculated according to the schema in Figure 3.

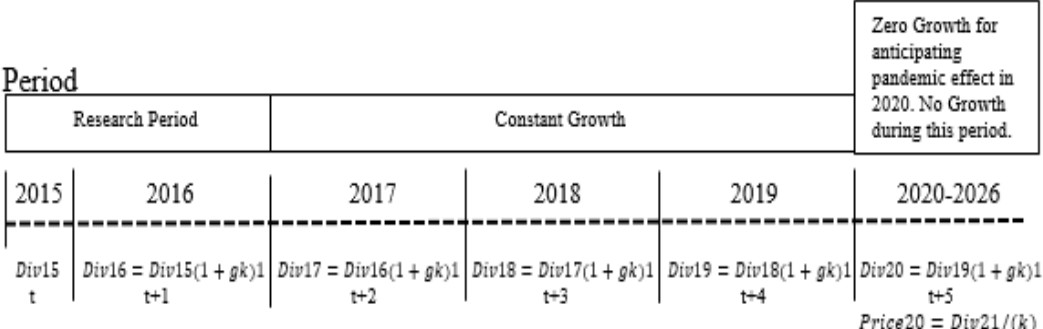

**Figure 3.** Stages of the estimated market price calculation for the period 2015–2020. Source: To be compiled from (Damodaran 2012), (Bodie et al. 2013), and (Ross et al. 2010).

This model was the adoption of the H model (Two-Stage Model for Growth) and has some stages in formulating the prediction model, as follows:

The first stage is for the prediction period of 2018-2019, where the value g = ROE × b, $g_k$ = average "growth" period 2013-2018, and k = free risk + beta (market return-free risk) as a proxy for calculating the present value of the future return. It refers to the CAPM model, which uses the indicator of expected return from the investor's viewpoint. The second stage for the prediction period of 2020–2022 uses assumptions such as Div20 = Div21 = Div22 = Div23 and *Price*20 = *Price* 21 = *Price*22 = *Price*23. The third stage for calculation is the market price, where the estimated price = Dividend Yield + Capital Gain (Bodie et al. 2013) and (Brigham and Houston 2013).

Based on some assumptions above in predicting the market price in the following period, the pattern of arranging the prediction model has been prepared for each period, where the systematic method can be seen on Table 1, as below:

**Table 1.** The mechanism of calculating the estimated price.

| Period | Prediction | Dividend Yield | Capital Gain |
|--------|------------|----------------|--------------|
| 2015 | Price Estimated at 2016 | $\dfrac{Div17}{(1+k1)^1} + \dfrac{Div18}{(1+k2)^2} + \dfrac{Div19}{(1+k3)^3} + \dfrac{Div20}{(1+k4)^4}$ | $\dfrac{Price20}{(1+k4)^4}$ |
| 2016 | Price Estimated at 2017 | $\dfrac{Div18}{(1+k1)^1} + \dfrac{Div19}{(1+k2)^2} + \dfrac{Div230}{(1+k)^3} + \dfrac{Div21}{(1+k4)^4}$ | $\dfrac{Price21}{(1+k4)^4}$ |
| 2017 | Price Estimated at 2018 | $\dfrac{Div19}{(1+k1)^1} + \dfrac{Div20}{(1+k2)^2} + \dfrac{Div21}{(1+k3)^3} + \dfrac{Div22}{(1+k4)^4}$ | $\dfrac{Price22}{(1+k4)^4}$ |
| 2018 | Price Estimated at 2019 | $\dfrac{Div20}{(1+k1)^1} + \dfrac{Div21}{(1+k2)^2} + \dfrac{Div22}{(1+k3)^3} + \dfrac{Div23}{(1+k4)^4}$ | $\dfrac{Price23}{(1+k4)^4}$ |
| 2019 | Price Estimated at 2020 | $\dfrac{Div21}{(1+k1)^1} + \dfrac{Div221}{(1+k2)^2} + \dfrac{Div23}{(1+k3)^3} + \dfrac{Div24}{(1+k4)^4}$ | $\dfrac{Price24}{(1+k4)^4}$ |
| 2020 | Price Estimated at 2021 | $\dfrac{Div22}{(1+k1)^1} + \dfrac{Div23}{(1+k2)^2} + \dfrac{Div24}{(1+k3)^3} + \dfrac{Div25}{(1+k4)^4}$ | $\dfrac{Price25}{(1+k4)^4}$ |

where: $k1$, $k2$, $k3$, $k4$ = expected return based on last five years. Obtaining $k1$ as an expected return, this model used CAPM from 2012 to 2016, where the free risk, beta, and free-market are calculated annually. Then $k2$ used the CAPM from 2013 to 2017 with an annual average of free risk, beta, and free-market, then the same treatment for $k3$ with CAPM from 2014 to 2018, the calculating $k4$ for dividend and price used the CAPM from 2014 to 2018.

The fourth stage for controlling the prediction model is performed by using the tracking signal as an indicator of the error range between estimated price t + 1 and average market price t + 1, the estimated price ranging between the prediction range −2.0 < Tracking Signal < 2.5 through indicators of cumulative forecast error and mean average deviation (Heizer et al. 2017), so that this calculation can be stated as high-accuracy prediction modelling.

Figure 3 depicts the new method for calculating the estimated price in the future, controlling this process using a tracking signal. Based on the study by Damodaran (2012), this research modifies the calculation of future firm value using the multistage growth model, whereby the H model combines constant and zero growth. The effect of COVID-19 was anticipated by assuming zero growth in the future, whereas constant growth was used for the previous year. Using the CAPM approach in calculating the expected return can be treated as a proxy of the present value of future return because the beta coefficient is a valid risk indicator. As a new approach, the validity of this measurement can be maintained with a low error in predicting the future price, thereby fulfilling the standard tolerance limit minimum. Considering the study by Heizer et al. (2017), which measures the error in estimating the future price as a function of tracking signal (−2 < tracking signal < 2.5), it can be stated that the H model has a high accuracy in predicting value. This study uses the tracking signal as an indicator of validity and accuracy in estimating dividends and market price in the following period.

### 3.2.2. Future Market Value Based on Equity

The future market value based on equity is derived from the yield book instrument model (Homer et al. 2013) when calculating the market value of bonds. As a new indicator for measuring prospects as a function of equity, a higher value of this ratio reflects a positive fluctuation in market price, which can be expressed as follows:

$$Future\ Market\ Value\ Based\ on\ Equity = \frac{Equity\ per\ share_{(t)}}{Estimated\ price_{(t+1)}} \tag{1}$$

### 3.2.3. Future Market Value Based on Earnings

The future market value based on earnings is a novel approach to measuring prospects, derived from the calculation of earnings yield instruments, developed by Wilcox (2007) and modified by Abraham et al. (2017) through the adjusted earning yield. A higher value of this ratio reflects a more positive perception, which can be expressed as follows:

$$Future\ Market\ Value\ Based\ on\ Earnings = \frac{Earnings\ per\ share_{(t)}}{Estimated\ price_{(t+1)}} \qquad (2)$$

### 3.2.4. Dividend Policy

The ability of dividend policy to reflect better prospects is a *debatable issue*; this policy has been a phenomenon in the capital market (Jabbouri 2016; Bassiouny et al. 2016; Mousa and Desoky 2019). Taleb (2012) stated that this policy harms leverage when the dividend payout ratio fulfills the expected return, thus reducing the capital cost. According to (Nekhili et al. 2016) who showed that this policy could moderate corporate governance, it was considered a moderating variable due to the high-yield dividend, which can be expressed as follows:

$$Dividend\ PayOut\ Ratio_{(t)} = \frac{Dividend\ Per\ Share_{re(t)}}{Earnigns\ Per\ Share_{(t-1)}} \qquad (3)$$

### 3.2.5. Discretionary Accrual Quality

Discretionary accruals are indicated by the residual error value ($\varepsilon_{j,t}$), whereby these accruals violate the accounting standards (Perotti and Wagenhofer 2014; Zarowin 2015; Kothari et al. 2016). This study used a model presented in the literature (Dopuch et al. 2012; Yoon et al. 2012; Zarowin 2015), which was shown to better estimate discretionary accruals when compared with that proposed by Kothari et al. (2005). It can be expressed as follows:

$$TA_{j,t} = \beta0 + \beta1A/R_{j,t} + \beta2A/P_{j,t} + \beta3INV_{j,t} + Profit\ margin_{j,t} + \varepsilon_{j,t} \qquad (4)$$

where $TA_{j,t}$ represents the total accruals for firm $j$ in period $t$, $A/R_{j,t}$ represents the account receivables for firm $j$ in period $t$, $A/P_{j,t}$ represents the account payables for firm $j$ in period $t$, $INV_{j,t}$ is the inventory for firm $j$ in period $t$, and $Profit\ margin_{j,t}$ represents the net earnings for firm $j$ in period $t$. The expectation for each variable is as follows: $\beta0 > 0$; $\beta1 > 0$; $\beta2 > 0$; $\beta3 > 0$; $\beta4 > 0$.

### 3.2.6. Tax Management

This study used the discretionary tax accrual quality as a proxy of tax management. These tax accruals do not violate tax regulations according to a residual error value ($\varepsilon_{j,t}$). This was a modification of the discretionary tax accruals model presented by Báez-Díaz and Alam (2012) and Choudhary et al. (2016) with the insertion of some new variables. Because tax management is a measurement of earning quality, discretionary tax accrual quality is calculated by multiplying discretionary tax accruals by $-1$. This model can be expressed as shown below.

Calculation of tax accruals:

$$TA_{j,t} = TTA_{j,t} + TBA_{j,t} \qquad (5)$$

where $TA_{j,t}$ represents the total accruals for firm $j$ in period $t$, $TTA_{j,t}$ represents the total tax accruals for firm $j$ in period $t$, and $TBA_{j,t}$ represents the total book accruals.

$$TTA_{j,t} = TI_{j,t} - CFO_{j,t} \qquad (6)$$

where $TTA_{j,t}$ represents the total tax accruals for firm $j$ in period $t$, $TI_{j,t}$ represents the taxable income for firm $j$ in period $t$, and $CFO_{j,t}$ represents the cash flow operational for firm $j$ in period $t$.

Calculation of discretionary tax accruals using total tax accruals:

$$TTA_{j,t} = (absolute)_{j,t} = \alpha_1 + \lambda_{11}(CFO)\lambda_{j,t} + \lambda_{12}(TL)_{j,t} + \lambda_{13}(SG)_{j,t} + \lambda_{14}(ANP)_{j,t} + \varepsilon_{j,t} \qquad (7)$$

where $TTA_{j,t}$ represents the total tax accruals for firm $j$ in period $t$, $CFO_{j,t}$ represents the cash flow operational for firm $j$ in period $t$, $TL_{j,t}$ is the tax liability for firm $j$ in period $t$, $SG_{j,t}$ is the sales growth for firm $j$ in period $t$, and $ANP_{j,t}$ is the adjusted net profit for firm $j$ in period $t$. The expectation for each variable is as follows: $\lambda_{11} > 0$; $\lambda_{12} > 0$; $\lambda_{13} > 0$; $\lambda_{14} > 0$. The basis for the measurement of control variables is presented in Table 2.

**Table 2.** The basis for the measurement of control variables.

| Variable | Measurement Formula | Scale |
|---|---|---|
| The dependent variable, as a measurement indicator of variable future market value | $The\ estimated\ price = \dfrac{Div_{t+1}}{(1+k1)^{t+1}} + \dfrac{Div_{t+2}}{(1+k2)^{t+2}} + \ldots + \dfrac{Div_{t+3}}{(1+k3)^{t+3}} + \dfrac{Price}{(1+k3)^{t+3}}$ <br> $Future\ market\ value\ on\ equity = \dfrac{Equity\ per\ share_{(t)}}{Estimated\ price_{(t+1)}}$ <br> $Future\ market\ value\ on\ earnings = \dfrac{Earnings_t}{Estimated\ price_{(t+1)}}$ | Ratio |
| Discretionary accrual quality = $(absolute\ \varepsilon_{j,t})\,X - 1$ | $Total\ accruals\ (TAC) = NI_{j,t} - CFO_{j,t}$ <br> $TAC_t = \beta0 + \beta1\dfrac{A}{R_{j,t}} + \beta2\dfrac{A}{P_{j,t}} + \beta3 INV_{j,t} + \beta4 Profit\ Margin_{j,t} + \varepsilon_{j,t}$ | Ratio |
| Tax management as a measurement indicator of discretionary tax accrual quality = $(absolute\ \varepsilon_{j,t})\,X - 1$ | $Adjusted\ net\ profit_{j,t} = Commercial\ net\ income_{j,t} + fiscal\ correction_{j,t}$ <br> $TTA_{j,t} = Adjusted\ net\ profit_{j,t} - cashflow\ operational_{j,t}$ <br> $DTA\ quality_{j,t} = TTA_{j,t} - (\alpha1 + \lambda11\,(cash\ flow\ operational)_{j,t}) + $ <br> $\lambda12\,(tax\ liability)_{j,t} + \lambda13(sales\ growth)_{j,t} + \lambda14\,(adjusted\ net\ profit)_{j,t}$ | Ratio |
| Moderating variable using dividend policy | $Dividend\ Pay\ Out\ Ratio_{(t)} = \dfrac{Dividend\ Per\ Share_{(t)}}{Earnings\ Per\ Sahre_{(t-1)}}$ <br> Measurement of period dividend policy $t$ and net income period $t-1$ | Ratio |
| First control variable (size) | $Book\ value = value\ of\ total\ assets\ for\ the\ period$ <br> $Size = \log\ natural\ (book\ value)$ | Ratio |
| Second control variable (sales growth) | $Total\ debt = short\text{-}term\ debt + long\text{-}term\ debt$ <br> $Debt\ to\ equity\ ratio = \dfrac{Total\ debt\ value}{Equity\ value}$ | Ratio |

## 4. Results

### 4.1. Descriptive Statistics

A summary of the descriptive statistics is provided in Table 3, where the normality distribution has been pointed out by comparing the smaller value of means than standard deviations. This descriptive testing is based on outlier testing, where the rejected data has amounted to 180 observations, and all observations have been 384. *Outlier testing* was achieved by applying the *Winsorize* model for data within $-1.5 >$ Z-score $> 1.5$ (Gujarati 2011); accordingly, we obtained 204 valid observations.

**Table 3.** Descriptive data.

| Description | N | Minimum | Maximum | Mean | Standard Deviation |
|---|---|---|---|---|---|
| Future market value based on equity | 204 | 0.234 | 0.821 | 0.419 | 0.017 |
| Future market value based on earnings | 204 | 0.161 | 0.736 | 0.293 | 0.143 |
| Discretionary accrual quality | 204 | −0.820 | 0.682 | 0.499 | 0.502 |
| Discretionary tax accrual quality | 204 | −0.960 | 0.890 | 0.426 | 0.620 |
| Dividend payout | 204 | 0.030 | 0.500 | 0.276 | 0.136 |
| Discretionary accrual quality × dividend | 204 | −0.246 | 0.951 | 0.752 | 0.561 |
| Discretionary tax accrual quality × dividend | 204 | −0.531 | 1.286 | 0.914 | 0.472 |
| Total assets (in thousands) | 204 | 476,149 | 1,447,865 | 859,497 | 63,617 |
| Growth of sales (in thousands) | 204 | −3.238 | 0.720 | −0.296 | 0.036 |
| Risk (beta) | 204 | 0.000 | 1.010 | 0.458 | 0.201 |

The descriptive statistics in Table 3 show the abnormal distribution of discretionary accrual quality and discretionary tax accrual quality. As preliminary statistical testing, the normality and heteroscedasticity have been run, as below.

The normality and heteroscedasticity testing in Table 4 pointed to the powerlessness of fulfilling this criterion, indicating many combination models publishing the accounting information quality. For the item of normality, this research used the extensive observation; where the sample >30, it can be assumed that it has the normality curve based on the *Central Limit theorem*; this formula can be seen as follows:

If X is the *mean* from one sample with size *n* taken from the population *mean* μ and *variance* $\alpha^2$, so the distribution has the limited function $Z = \frac{\bar{x}-\mu}{\sigma/\sqrt{n}}$; it is too close to the normal distribution when $n \to \infty$. The mathematic formulation can be written, as follows:

$$\lim_{n\to\infty} P(Z \le \pi) = \frac{1}{\sqrt{2\pi}} \int_{-\infty}^{x} e^{-\frac{y2}{2}} dy \tag{8}$$

Lebert (2019) stated that accrual quality in the capital market has an abnormal distribution because there are many variations in earning quality and tax management, in addition to the consequences of using panel data combining cross-sectional and time-series data (Hair et al. 2010). When using panel data, there is an imbalance across years because dividend policy cannot be implemented every year; it is recognized as a regular corporate action. Thus, panel data have a high degree of freedom in determining the most efficient econometric model by gathering more information compared to cross-sectional and time-series data.

**Table 4.** Normality and heteroscedasticity.

| Description Variables | KS (*) | Sig | Note |
|---|---|---|---|
| Discretionary accrual quality ($X_{DAQ}$) | 0.043 | 0.186 | Ab and Hetero |
| Discretionary tax accrual quality ($X_{DTQ}$) | 0.029 | 0.448 | Ab and Hetero |
| Future market value based on equity | 0.037 | (**) | Abnormal |
| Future market value based on earnings | 0.016 | (**) | Abnormal |
| Dividend payout ($X_{Div}$) | 0.953 | 0.007 | Normal and Homo |
| Discretionary accrual quality × dividend ($X_{MDiv1}$) | 0.017 | 0.179 | Ab and Hetero |
| Discretionary tax accrual quality × dividend ($X_{MDiv2}$) | 0.002 | 0.427 | Ab and Hetero |
| Log total asset ($X_{TA}$) | 0.751 | 0.019 | Normal and Homo |
| Growth sales ($X_{Sa}$) | 0.496 | 0.043 | Normal and Homo |
| Risk ($X_{Ri}$) | 0.614 | 0.035 | Normal and Homo |

Note: Using Kolmogorov Smirnov (*) for normality testing, where KS > 0.05 as normality. Using residual as independent with Glejser Testing for heteroscedasticity, where sig < 0.05 as homogeneity. Ab and Hetero = abnormal and heteroscedasticity. Normal and Homo = normal and homogeneity. (**) = It is not tested because of the role as independent variable.

*4.2. Statistical Testing*

This study implemented a one-tailed statistical analysis because of the one-way hypothesis. The statistical testing of panel data is summarized in Table 5.

According to Table 5, the first model was a random effect model, while the second model was a fixed-effect model; this indicated that the item of Constanta in the regression model is a meaningless indicator. Therefore, there was no need to fulfill a classic testing model because of the highly dispersed distribution data (Salleh et al. 2011; Alipour et al. 2019). A summary of statistical testing, along with the regression model, is presented in Table 6.

**Table 5.** The summary of panel data testing.

| Phase-In Testing Data Panel | FMV Based on Equity First Model | FMV Based on Earnings Second Model |
|---|---|---|
| Chow testing | $p$ value = 0.083 (>0.05) $H_0$ accepted Common effect model | $p$ value = 0.006 (<0.05) $H_0$ rejected Fixed effect model |
| Hausman testing | Not applied | $p$ value = 0.004 (<0.05) $H_0$ rejected Fixed effect model |
| Lagrange multiplier testing | Prob. Breusch–Pagan = 0.093 (>0.05) $H_0$ rejected Fixed effect model | Not applied |

**Table 6.** The First regression testing in this model.

| Description of Variable | Future Market Value-Based on Equity | | | | Future Market Value-Based on Earnings | | |
|---|---|---|---|---|---|---|---|
| | Coefficient | t | Sig(*) | Hypothesis | Coefficient | t | Sig(*) |
| Constant | −0.871 | −0.782 | 0.089 | | −0.792 | −0.632 | 0.296 |
| **Dependent variables** | | | | | | | |
| Discretionary accrual quality ($X_{DAQ}$) | 0.127 | 4.181 | 0.007 | H1a and H1b accepted | 0.091 | 4.792 | 0.009 |
| Discretionary tax accrual quality ($X_{DTQ}$) | 0.108 | 2.413 | 0.028 | H2a and H2b accepted | 0.228 | 1.892 | 0.031 |
| Dividend payout ($X_{Div}$) | 0.274 | 5.041 | 0.004 | | 0.239 | 4.812 | 0.007 |
| Discretionary accrual quality × dividend ($X_{MDiv1}$) | 0.313 | 3.424 | 0.018 | H3a and H3b accepted | 0.205 | 2.819 | 0.021 |
| Discretionary tax accrual quality × dividend ($X_{MDiv2}$) | −0.051 | −0.763 | 0.197 | H4a and H4b rejected | −0.037 | −0.926 | 0.231 |
| **Control variables** | | | | Note | | | |
| Log total asset ($X_{TA}$) | 0.071 | 2.859 | 0.017 | Significant | 0.058 | 2.042 | 0.032 |
| Growth sales ($X_{Sa}$) | 0.114 | 5.581 | 0.002 | Significant | 0.179 | 4.259 | 0.007 |
| Risk ($X_{Ri}$) | −0.373 | −2.085 | 0.023 | Significant | −0.486 | −1.664 | 0.038 |
| Analysis of variance F-test calculated Sig. level Adjusted $R$-square Coefficient $R$-square Durbin Watson value | 8.341 (>$F_{Table}$ 0.338) 0.0000 (<0.05) 0.245 0.301 2.087 (1.845 < X < 2.154) | | | | 3.872 (>$F_{Table}$ 0.338) 0.0000 (<0.05) 0.206 0.274 1.965 (1.845 < X < 2.154) | | |

Note: Sig(*) = one-tailed significance (Sig/2); F calculated = (0.05, 0.338); t calculated = (0.05, 0.519).

According to the *t*-test in Table 6, the independent variable had a stimulating effect on the dependent variables. According to the F-test, the model had high significance; thus, it can be used for predictive modeling and supported by t testing as an indicator of partial relationship.

First regression model:

*Future market value on equity*
$$= -0.871 + 0.127 XDAQ_1 + 0.108\ XDTQ + 0.274 XDiv + 0.313\ XMDiv1 - 0.051 XMDiv2 + 0.071 XTA \quad (9)$$
$$+0.114 XSa - 0.373\ XRi.$$

The discretionary accrual quality had a significance level of 0.007 (<0.05 (error level)) and a positive coefficient of regression (0.127); thus, hypothesis H1 could be accepted. The discretionary tax accrual quality had a significance level of 0.028 (<0.05 (error level)) and a positive coefficient of (0.108); thus, hypothesis H3 could be accepted. The moderation between discretionary accrual quality and dividend policy had a significance level of 0.018(<0.05 (error level)) and a positive coefficient of regression (0.313); thus, hypothesis H5

could be accepted. The moderation between discretionary tax accrual quality and dividend policy had a significance level of 0.097 (>0.05 (error level)) and a negative coefficient of regression (−0.051); thus, hypothesis H7 could be rejected.

Second regression model:

*Future market value on earnngs*
$$= -0.792 + 0.091\text{XDAQ} + 0.228\text{XDTQ} + 0.239\text{XDiv} + 0.205\text{XMDiv1} - 0.037\text{XMDiv2} + 0.058\text{XTA} \quad (10)$$
$$-0.179\text{XSa} - 0.3486\text{XRi}.$$

The discretionary accrual quality had a significance level of 0.009 (<0.05 (error level)) and a positive coefficient of regression (0.091); thus, hypothesis H2 could be accepted. The discretionary tax accrual quality had a significance level of 0.031 (<0.05 (error level)) and a positive coefficient of regression (0.228); thus, hypothesis H4 could be accepted. The moderation between discretionary accrual quality and dividend policy had a significance level of 0.021 (<0.05 (error level)) and a positive coefficient of regression (0.205); thus, hypothesis H6 could be accepted. The moderation between discretionary tax accrual quality and dividend policy had a significance level of 0.231 (>0.05 (error level)) and a negative coefficient of regression (−0.037); thus, hypothesis H8 could be rejected.

After re-testing the regression model without the insignificant variable, the regression testing can be arranged systematically, as below:

First regression model: without using the insignificant variable:

*Future market value on equity*
$$= -0.041 + 0.201\text{XDAQ} + 0.117\text{XDTQ} + 0.296\text{XDiv} + 0.417\text{XMDiv1} + 0.091\text{XTA} + 0.197\text{XSa} - 0.461\text{XRi}. \quad (11)$$

Second regression model: without using the insignificant variable:

*Future market value on earnngs*
$$= -0.092 + 0.137\text{XDAQ} + 0.294\text{XDTQ} + 0.262\text{XDiv} + 0.368\text{XMDiv1} + 0.083\text{XTA} - 0.284\text{XSa} - 0.509\text{XRi}. \quad (12)$$

Based on Table 7, the re-testing without using the insignificant variable shows that the more significant coefficient regression of all dependent and control variables on the independent variable illustrated the more sensitive effect of significant variables, which can be detected by some critical statistical tools, such as t and F testing. It is related to the other measurement of the multiple regression's fitting tests, such as the adjusted R square and variance F testing, which means that this model could be used to predict the management performance in the future with high accuracy. The t testing indicated that the dependent variable has a partial relationship with independent variables as the precondition of this predictive modeling, which could be used for forecasting with high accuracy. The effectiveness rate of the dividend payout ratio as a signaling effect of the high obedience to accounting standards and tax regulation has been high because of management's strong willingness to gain a high probability of better prospects. The high earning quality has played a crucial role in obtaining the investor's positive perception when the positive movement of market price stimulates "good news" for a better return in the future, which it refers to be protective information of opportunity behavior.

**Table 7.** The second regression testing in this model.

| Description of Variable | Future Market Value Based on Equity | | | Future Market Value Based on Earnings | | |
|---|---|---|---|---|---|---|
| | Coefficient | t | Sig(*) | Coefficient | t | Sig(*) |
| Constant | −0.041 | −0.982 | 0.101 | −0.092 | −0.771 | 0.383 |
| **Dependent variables** | | | | | | |
| Discretionary accrual quality ($X_{DAQ}$) | 0.201 | 5.981 | 0.004 | 0.137 | 5.483 | 0.005 |
| Discretionary tax accrual quality ($X_{DTQ}$) | 0.117 | 3.123 | 0.011 | 0.294 | 3.194 | 0.019 |
| Dividend payout ($X_{Div}$) | 0.296 | 6.147 | 0.003 | 0.262 | 4.984 | 0.006 |
| Discretionary accrual quality × dividend ($X_{MDiv1}$) | 0.417 | 4.134 | 0.009 | 0.368 | 3.019 | 0.014 |
| **Control variables** | | | | | | |
| Log total asset ($X_{TA}$) | 0.091 | 4.715 | 0.008 | 0.083 | 2.331 | 0.016 |
| Growth sales ($X_{Sa}$) | 0.197 | 6.912 | 0.000 | 0.284 | 6.262 | 0.002 |
| Risk ($X_{Ri}$) | −0.461 | −4.326 | 0.007 | −0.509 | −3.457 | 0.013 |
| Analysis of variance F-test calculated | 11.435 (>$F_{Table}$ 0.271) | | | 5.564 (>$F_{Table}$ 0.271) | | |
| Sig. level | 0.0000 (<0.05) | | | 0.0000 (<0.05) | | |
| Adjusted *R*-square | 0.297 | | | 0.247 | | |
| Coefficient *R*-square | 0.342 | | | 0.304 | | |
| Durbin Watson value | 2.095 (1.845 < X < 2.154) | | | 2.001 (1.845 < X < 2.154) | | |

Note: Sig(*) = one-tailed significance (Sig/2); F calculated = (0.05, 0.271); t calculated = (0.05, 519).

By separating the statistical testing to FMV based on equity and earnings, the following critical points could be identified:

1.  The regression coefficient revealed that a high obedience level to accounting standards positively affects discretionary accrual quality in FMV, as a function of both earnings and equity. This was supported by the moderating variables, suggesting that the dividend push management toward a high obedience level to accounting standards.

2.  High compliance with tax regulations played a primary role in increasing the accuracy of predicting future returns, as indicated by the effect of discretionary tax accrual quality on FMV as a function of equity and earnings. The moderation of the dividend revealed a different result as the dividend did not strengthen the effect of tax management on FMV as a function of equity and earnings.

3.  Sales and total assets as control variables had a significant positive contribution to future market value as a function of both equity and earnings. This suggests actual earnings as an indicator of a high probability of better prospects for investors.

4.  Risk as a control variable had a significant negative impact on future market value as a function of equity and earnings. This indicates a relationship between obedience and agency cost. Thus, opportunistic behavior increases the uncertainty and unpredictability of future earnings.

## 5. Discussion

Statistical testing revealed that high earnings and good tax management are positive signals for investors when predicting prospects (notably, the required return rate). Dividend policy plays a critical role in increasing the trust of investors when management has a low capital cost. This study found a trend in market price fluctuation, whereby dividend policy minimizes opportunistic behavior through the higher involvement of investors. This is supported by Mousa and Desoky (2019) and Pathak and Ranajee (2020), who stated that this policy reduces misleading information to a minimum because of high shareholder involvement. When the dividend payout ratio fulfills the expected return, the firm exhibits better performance (Deng et al. 2017), as this is a sign of a low manipulation of earnings. Positive fluctuation occurs in the presence of good signals, whereas unfavorable fluctuation leads to a negative market value, which will handicap management in reaching better performance in the future.

Discretionary tax accrual quality positively contributes to the perception of investors, but this is subject to moderating variables. Management should aim to simultaneously

improve obedience to accounting standards and tax regulations; however, this is tricky because of the liquidity of cash flow, as supported by Yorke et al. (2016), Lee (2016), and Jacob and Schütt (2020). This study determined that management has a proclivity to implement a high-yield dividend policy to send a good signal while ignoring compliance with tax regulations. This allows for minimizing the intensity of internal conflict and gaining low capital costs.

In line with Datta et al. (2013), when there is an incapability of maintaining the business in the long term, the management has the proclivity to make any violation a sign of low earnings quality and then is aimed for disseminating the tracking on the right track as an opportunity behavior. Empirically, the statistical testing in this study found a similar connection between high-earning accruals and investor perception. This result implies a signaling effect whereby management tends to send "good news" about the company. The same opinion has been strengthened by Wu et al. (2016) by accentuating that the earning quality indicates management's ability to obtain better prospects, as it can indicate significant business growth. Referring to this behavior for each party, Askari et al. (2019) have categorized this linking as an implication of game theory, while Chen and Wu (2021) emphasize that the critical role of the investor is in making the investment decision based on the accounting information in predicting a better prospect in future. Therefore, the relationship between high-quality accounting information and investor action can be considered interactive feedback, with one dynamically stimulating the other. The implication of game theory in publishing high earning quality has been new literature on earning quality. When there is no tolerance for any violation, this negative perception is indicated by the negative movement of the market price. With the high validity in testing the impact of earnings quality, this relationship has been an answer for smoothing the high fluctuation of market price when the accounting information has been used to take the rational decision model. This research aims to deduct the probability of speculative motive as bounded rationality decision modeling. As a strategic benefit of the investment in the capital market as an attractive investment, (Eldomiaty et al. 2020) and (Wang and Li 2020) have proven this positive relation between the future return and the capability of manufacturing companies in reaching full employment. It refers to an ideal condition of the economic macro. This research has empirically proven that the relationship between investors and management positively affects high earning quality on market price movement.

This study provides a decision tree model (Figure 4) for mapping the connection between high-quality accounting information and market value in the following period, where the probability of positive investor perception can be estimated using Bayes' theorem (Kaplan 1996; Hutton and Stocken 2021).

Figure 4 reveals the applicability of the game theory in publishing high-quality accounting information. The investor is alerted of accruals and prioritizes a high obedience to accounting standards and compliance with tax regulations as fundamental indicators of high-quality financial reporting. According to statistical testing, there is no significant effect of dividend policy and tax management on future market value; this supports the free cash flow hypothesis, where management has the proclivity to reduce the intensity of internal conflict to attain better performance. Dividend plays a role in improving obedience to accounting standards, with a high-yield dividend indicating a high probability of gaining better prospects, thereby stimulating the positive movement of the market price. This decision tree model confirms the critical role of dividend policy in achieving high obedience, which illustrates honest earnings or actual performance. The mechanism of market price movement can be described as follows:

1. A "positive market value" better indicates a high potential of fulfilling the future expected (bold line); this represents the positive movement of market price and low fluctuation because real earnings are a sign of obtaining a better future return.

2. A "negative market value" indicates a low potential of fulfilling the future expected return (thin line); this represents the negative movement of market price and volatile fluctuation due to unsatisfactory performance.

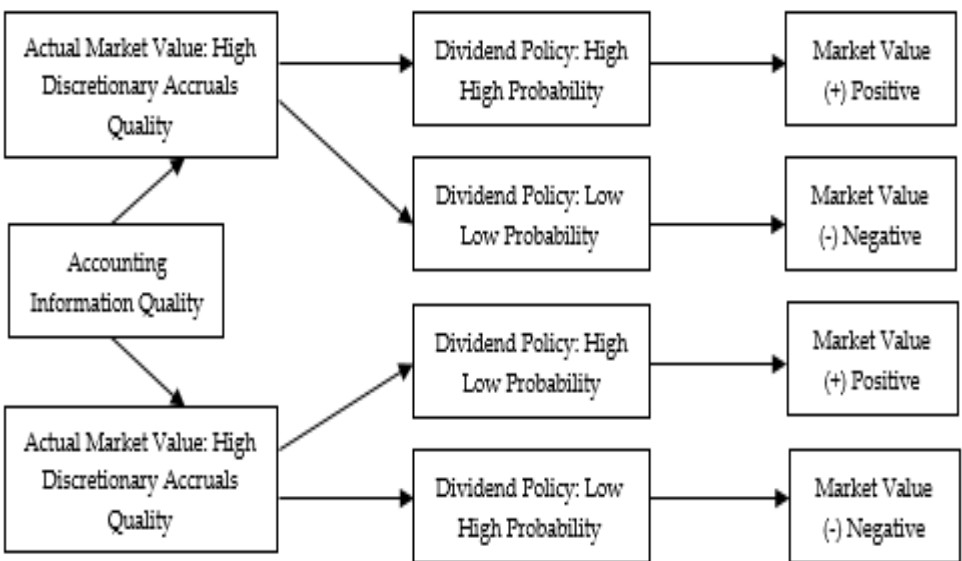

**Figure 4.** Decision tree model. Note: the researchers' data.

After mapping the decision tree model, the probability of firm value was calculated according to Bayes' theorem as a predictive model for estimating investor action. The calculation model can be distinguished into patterns of positive and negative movement of the market price as expressed below. The first predictive model for a positive market value to estimate the probability of taking a buy position is as follows:

$$P(HighDiv|DAQ,\ High) = \frac{P(DAQ,\ High|HighDiv).\ P(DAQ,\ High)}{P(DAQ,\ High|HighDiv).\ P(DAQ,\ High) + P(DAQ,\ High|LowDiv).\ P(LowDiv)} \quad (13)$$

where $P(DAQ,\ High)$ is the probability of a firm having a high discretionary accrual quality. $P(HighDiv)$ is the probability of the dividend policy featuring a high payout ratio, $P(LowDiv)$ is the probability of the dividend policy featuring a low payout ratio, $P(DAQ,\ High|HighDiv)$ is the probability of high discretionary accrual quality when a high-yield dividend has been implemented, $P(DAQ,\ High|LowDiv)$ is the probability of high discretionary accrual quality when a low-yield dividend has been implemented, and $P(HighDiv|DAQ,\ High)$ is the probability of a high-yield dividend being implemented in a firm with a high discretionary accrual quality.

The second predictive model for a negative market value to estimate the probability of taking a sell position is as follows:

$$P(HighDiv|DAQ,\ Low) = \frac{P(DAQ,\ Low|HighDiv).\ P(DAQ,\ Low)}{P(DAQ,\ Low|HighDiv).\ P(DAQ,\ Low) + P(DAQ,\ Low|LowDiv).\ P(LowDiv)} \quad (14)$$

where $P(DAQ,\ Low)$ is the probability of a firm having a low discretionary accrual quality. $P(HighDiv)$ is the probability of the dividend policy featuring a high payout ratio, $P(LowDiv)$ is the probability of the dividend policy featuring a low payout ratio, $P(DAQ,\ Low|HighDiv)$ is the probability of low discretionary accrual quality when a high-yield dividend has been implemented, $P(DAQ,\ Low|LowDiv)$ is the probability of low discretionary accrual quality when a low-yield dividend has been implemented, and $P(HighDiv|DAQ,\ Low)$ is the probability of a high-yield dividend being implemented in a firm with a low discretionary accrual quality.

By mapping the feedback between investors and management, some practical implications of this research are presented below.

1.  The regulator in the capital market is provided with feedback to release the regulation, thus giving management an attractive incentive to publish high-quality financial reporting. The regulation should boost investors' trust as a function of high-quality financial reporting when a legal standing approach has been needed to deduct a high level of accruals as a sign of any infringement.

2.  Considering the impact of implementing a high-yield dividend, the authoritative regulator must design the preventive effect of this corporate policy by creating some barriers for this dividend policy model, as it generates the high involvement of dominant shareholders, which is a violation of the minority investor.

3.  The regulator should release the alluring rule to force management to run this dividend policy regularly as a standard corporate policy. Unquestionably, there is no tax on the dividend. Regulators in the capital market should design an appealing controlling and monitoring model for the company to annually and consistently implement a dividend policy model. A high-yield dividend should be intercepted quantitatively by the "flawless" approaches, representing a new obstacle in managing the liquidity of cash flow when funding expanded business activity; this is a widely opened chance for opportunistic behavior.

## 6. Conclusions

The result of this study reveals that an implication of game theory, earning quality, and tax management significantly contribute positively to future market value when tested directly without using moderating variables. This illustrates that the investor has the primary requirement of obtaining high-quality financial reporting as a sign of fulfilling the expected future return, backed up by the growth of total assets and sales, as well as low risk. When management has high compliance with accounting standards and tax regulations, it significantly stimulates the positive movement of the market price. High earning quality would pave the way for the investor to predict prospects with high accuracy because of the low probability of creating a crucial problem with the government's regulatory party.

The moderation between discretionary accrual quality and dividend policy indicated that a dividend policy reduces the chance of opportunistic behavior and stimulates high investor involvement in controlling strategic advantages. This is related to the powerful force of management complying with accounting standards with minimum opportunistic behavior, which is favorable for accruals. Empirically, the dividend policy does not have a positive contribution to tax management and future market value, whereby management has the choice of obtaining a low capital cost by implementing a high-yield dividend. According to the free cash flow hypothesis, the willingness to disseminate the signal of a firm being on the right track is a priority compared with being legally compliant with tax regulations.

The impact of dividend policy has broadened the enrichment literature on earnings management; when the payout ratio has matched the expected return, it will have stimulated the high involvement of shareholders in making the strategic decision because maintaining the firm value as a protective action covers up the attention of a high or low probability of tax investigation. The high involvement of shareholders has been a sign of good prospects, which is used to force the management to be prudent in designing the accounting treatment policy. By underlining management's willingness to implement high yield management, this one pointed out the magnitude effect of accounting information. It is inclusion of the discretionary accrual has always existed, so it needed strict attention for learning over this chance because of a low sustainable existing business prospect. By illustrating the positive contribution to testing the accounting information, this research implies that high earnings quality has been a predictor of good prospects at a high accuracy level.

This study recommends that the authorities in Indonesia's capital market take a solid stance on management releasing high-quality financial reporting, including mandatory

dividend policy as well as tax incentives. Because of the trickle-down effect, the dividend policy allows the investor to predict future returns with high accuracy, which simulates the high involvement of shareholders in controlling accounting policies. Lastly, the accounting information has the structural function of protecting the investor from investing in a secured area. Therefore, the regulator should aim to continuously improve control over high-quality financial reporting, mainly using a punishment and reward system, thus encouraging high compliance with accounting standards and tax regulations. Investing in the capital market becomes attractive when there is a mutual relationship between investors and management toward reaching full employment.

## 7. Limitations and Future Research Directions

This study had a few limitations. Firstly, a constant growth rate was considered in the calculation of the indicator g (growth) over the next 5 year period. Secondly, secondary data were used, which led to highly dispersed variation. Thus, statistical testing involved a large dataset, with 180 samples being rejected and 384 samples being collected for observation. Lastly, because of the COVID-19 pandemic in 2020, the statistical testing assumed zero dividend growth for the subsequent 5 years to capture the actual estimation of prospects. This refers to the extended period required for the recovery of supply and demand, as well as the probability of a global economic recession.

For future research, the modified model in measuring future market value as a proxy for the perception of investors in the following period has been needed to be testing the future performance, and this future model could use multiple growth rates (e.g., high, stable, and zero growth) for the valuation of firm value. Based on the firm valuation, the cash flow will have been considered a variable to predict the future market value when the investors do not have the fundamental attention to the profit for the current situation. It is a consequence of the COVID-19 pandemic when a high probability of prospects in the future can be obtained. The uniqueness of future market value is the most appropriate characteristic for a representative sign of accurately predicting in the short term, for which the manufacturing is hard-pressed for reaching the economic scale; this should be developed into an advanced framework modeling for predicting the medium period. There is one suggestion for using cash flow operation for firm valuation, where this variable reflects the capability of managing a sustainable business in the long run. This one aims to smooth market price fluctuation, particularly speculative motives. As the item of predictability, predicting a short–medium range period will be a guaranteed sign-on investing in the capital market, and it refers to a "good sign". Furthermore, researchers should be aware of several limitations of beta instruments when measuring risk levels; thus, different approaches should be considered. A more comprehensive calculation of risk and estimation of the market price is required to detect the company's existence in the long run, for example, using the arbitrage pricing theory (APT) model.

**Author Contributions:** M.S. is responsible for conceptualization, methodology, and formal analysis. Y.A. is responsible for writing-review and editing, data analysis and curation. I.D. is responsible for data validation and project administration M.S., Y.A. and I.D. have finished all writing processes, including origins draft preparation and review and editing. We declare that all authors have equal contributions to this research All authors read and approved the final manuscript.

**Funding:** This research has no funding affiliation, so this publication has an independent analysis. This APC is financed by private, where each author has the same proportion.

**Institutional Review Board Statement:** Not applicable.

**Informed Consent Statement:** All data in this paper used all the published data, where there is no identifiable private information in collecting data. Informed consent was obtained from all participants personally involved in this research.

**Data Availability Statement:** All processing of raw data was generated at the IKPIA Perbanas Institute and Universitas Binaniaga Indonesia. All data supporting this study are available from the corresponding author by proposing a request.

**Conflicts of Interest:** There are no conflict of interest to declare relevant to this article's content, where the authors are in an independent position. There is no authors' involvement in any listed firms' structure, and we are academic researchers at our university.

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
