# Peer review of "The Relationship between Dividend Policy and Earnings Quality: The Role of Accounting Information in Indonesia’s Capital Market"

_economies, doi:10.3390/economies10060140_

Round 1
Reviewer 1 Report
This paper studies the influence of accounting information in the Indonesia
capital market. The paper is in general well written. I have some minor comments for the authors to consider.
1) In the introduction, you may want to detail your motivation on working with the Indonesian market. For instance, why is this market worth investigating?
2) It seems the the robustness of the tested hypotheses is not demonstrated. Have you considered using some alternative and popular methods (used in the literature) to evaluate the sensitivity of your baseline results?
3) Some presentation could be improved. For example, on page 8, quite a few equations are displayed for estimated prices. Those are quite similar and may be presented in a matrix form to save space.
4) There are many coloured chunks in this paper, which seems to be unnecessary.
Reviewer 2 Report
This is an interesting study of dividend policy and earnings, finding dividend policy is important. I have the following comments:
- In the introduction the authors need to emphasise the original contribution of this paper to the literature.
- The data needs a bit more discussion, on line 279 it notes there are 384 observations, but in Table 2 it notes there are 204, is the difference due to missing observations? In which case was the panel balanced? Also why were only firms with a positive growth rate included?
- Why have the authors conducted one-tailed tests rather than two tailed? I would have thought a two tailed test would be more appropriate here.
- Were robust standard errors used to remove the heteroskedasticity?
- The title refers to a dynamic relationship, but the dynamics (lagged variables) are not obvious. If the focus of the paper is to be dynamic it needs to use a dynamic panel model, such as the Arellano-Bond model, otherwise the term dynamic should be removed from the title.
Minor points:
- There are some parts of the manuscript that are shaded, this needs removing or explained.
- It would be helpful for the variables to refer to what they represent in some form rather than all Xs in the equations, for instance tax could be 't'.
-As it is panel data, the subscript on the variables should be 'it'
- Although well written the paper could do with a final proof reading. For instance I am not sure about line 504: 'this is considered to underline this fittest predictive modeling'
Round 2
Reviewer 2 Report
The revisions have been done well, a couple of minor points, I don't think using a panel data regression will remove the heteroskedasticity, unless you use the robust standard errors such as Whites standard errors. Most software does this easily, in Stata you just need to add the word 'robust' after the variables and they are automatically adjusted to remove the heteroskadasticity.
The 'it' referred to the subscript, i.e. Xit.
Author Response
Please see the attachment.

This manuscript is a resubmission of an earlier submission. The following is a list of the peer review reports and author responses from that submission.
Round 1
Reviewer 1 Report
#1 The author must consider whether the fixed discount rate is in line with the actual environment. In order to simplify the present value reflected by the time value of money, the general textbook will explain the value of money with a fixed discount rate. But the author defines k in this paper as follows: " dk = free risk + beta (market return-free risk) as a 299 proxy of calculating the present value of future return.".
(1) risk free rate as discount rate
(2) beta
Are these two data fixed every year? Basically, the risk free rate is not necessarily fixed, and the beta is also not fixed. If there are major changes in the market environment, these values may be Makes changes, although defined as risk-fee. In addition, why the discount rate k can be defined in this way?
#2 The author suddenly presents a decision tree model without any connection to the research topic. How does such a model relate to the content of the topic you discussed earlier? The author must explain the connection and must avoid over-extending the results of the topic.
#3 The future market is a short-term derivative financial product. Whether a factor of relatively long-term financial report analysis such as "The Impact of Dividend Policy on Earnings Quality and Tax Management" can truly reflect the value of the futures market is debatable.
Reviewer 2 Report
I read the paper. I am satisfied with their revision and I find the quality of the paper is satisfactory. Thus, I recommend accepting their paper.
Reviewer 3 Report
The authors made a great effort to improve the text. In my opinion it is not very advanced but correct and could be published in present form.
Reviewer 4 Report
Accepted in present form
Round 2
Reviewer 1 Report
#1 I don't think the author is very clear about what the discount rate of NPV is. The textbook writing method will be a fixed discount rate for each period. When the author considers risks, the discount rate must be calculated according to the estimated risk premium in the current environment. Adjustment, especially in the capital market (futures market) facing high risk, it is impossible to represent the discount rate with a fixed value in the long-term discount rate.
Simple two-period description. The first period discount rate is k1, and the second period discount rate is k2, so NPV= cash flow 1/(1+k1)+cash folw 2/[(1+k1)(1+k2 )], instead of the author's NPV=cash flow1/(1+k)+cash flow 2/[(1+k)(1+k)].
#2 Well done than the prtevious version.
#3 The author changed the title of the title, and the content is still the discussion of future market. Whether it is consistent with the research content needs to be further verified by the author. Basically, the theory and practice are related to future market issues. If the author wants to introduce a long-term equilibrium empirical model, basically With the future market in the market (whether efficient or not?), the author needs to consider the characteristics of the short-term future market, so the discussion of such issues may be of little or unrealistic reference value from the perspective of investors in practice.